# Evaluation of the Intrinsic and Perceived Quality of Sangiovese Wines from California and Italy

**DOI:** 10.3390/foods9081088

**Published:** 2020-08-10

**Authors:** Valentina Canuti, Annegret Cantu, Monica Picchi, Larry A. Lerno, Courtney K. Tanabe, Bruno Zanoni, Hildegarde Heymann, Susan E. Ebeler

**Affiliations:** 1Department of Agricultural, Food, Environmental and Forestry Sciences and Technologies (DAGRI), University of Florence, via Donizetti 6, 50144 Firenze, Italy; monica.picchi@unifi.it (M.P.); bruno.zanoni@unifi.it (B.Z.); 2Department of Viticulture and Enology and The Food Safety and Measurement Facility, University of California, One Shields Avenue, Davis, CA 95616, USA; acantu@ucdavis.edu (A.C.); lalerno@ucdavis.edu (L.A.L.); cktanabe@ucdavis.edu (C.K.T.); hheymann@ucdavis.edu (H.H.); seebeler@ucdavis.edu (S.E.E.)

**Keywords:** Sangiovese, wine regionality, intrinsic quality, perceived quality, sensory profile, volatile profile, polyphenol composition, typicality

## Abstract

Sangiovese is the most cultivated red grape variety in Italy where it is certified for the production of several Protected Designation of Origin (PDO) wines, and it is one of the most cultivated Italian red grape varieties in California. Despite the global distribution of this variety, there is a lack of international studies on Sangiovese grapes and wines. For this reason, the present study aimed to compare 20 commercial Sangiovese wines from 2017 harvest, 9 produced in Italy (Tuscany) and 11 in California, in order to evaluate the intrinsic and perceived quality. The eligibility, identity, and style properties (the intrinsic quality) of the wines were evaluated. A group of 11 Italian experts evaluated the perceived quality by rating the typicality of the wines. The experimental data showed that the intrinsic quality of Sangiovese wine samples was affected by the growing area; in particular, the wine resulted very different for the color indices and polyphenol composition. The above differences in intrinsic quality levels did not lead to a different evaluation of the perceived quality (typicality) by the wine experts. The results evidenced that Sangiovese variety is recognizable also if grown outside its original terroir, and fresh and fruity wines were considered more typical. This study expands our current knowledge of Sangiovese wines and the contribution of regional characteristics to the composition of wine.

## 1. Introduction

Sangiovese is the most important Italian red grape variety, with origins in Tuscany and Calabria in the south of Italy [1,2]. Through time, Sangiovese has always been considered as a good-quality cultivar for wine production. Today, it is the basis of internationally known DOC (Denominazione di Origine Controllata) and DOCG (Denominazione di Origine Controllata e Garantita) wines from Tuscany in Italy, the region where it is most cultivated. Sangiovese is also grown in Argentina, California, France, and in other countries such Australia and Chile, but to a lesser extent. Currently, Sangiovese is also the cultivar with the highest number of clones registered in the Italian National Catalogue of Grapevine Cultivars [3] showing 130 clones in 2020.

Despite its global distribution and importance, there are a limited number of international studies on Sangiovese grapes and wines. The most substantial scientific works were presented at the International Symposium of Sangiovese [4,5] between 2000 and 2004 in Italy. Some of these studies showed the influence of regionality and terroir on the flavor profile of Sangiovese wine.

Recent research has focused on the relationship between Sangiovese clones’ grape quality and their oenological properties [6] and polyphenol composition of Sangiovese wines and the mouthfeel perception [7,8]. A first study about the chemical characterization and comparison of Sangiovese wines from California and Italy was recently reported [9]. The findings demonstrated that the California Sangiovese wines have common characteristics with the Italian wines, particularly related to some volatile grape-derived compounds. These components can be associated with the varietal character of Sangiovese in both regions as the wines in each region retain a common set of characteristics, particularly for the varietal volatiles that originated from grape.

There have been numerous studies characterizing regional chemical and sensory differences in wines. Typical characterization of wines includes relationship between sensory profile and volatile compounds [10,11,12], polyphenols [13,14,15,16], and elements [17,18].

A much smaller number of studies have compared the chemical and sensory profiles of wines from multiple countries, including Malbec from Argentina and California [19,20], red wines (Shiraz, Cabernet Sauvignon, Merlot and blends, including Grenache and Viognier components) from Australia and China [21], and Sauvignon blanc wines from France, New Zealand, Spain, South Africa, and the United States [22,23].

Regional typicality is an important concept for the wine industry as it not only delineates geographic areas but also comprises wines with recognizable sensory characteristics and composition [24,25,26].

Characterizing regional differences in wines requires studying the intrinsic quality as its inherent physical-chemical characteristics. Moreover, perceived quality is rated by experts, critics, and consumers, which altogether defines a profile that describes common regional characteristics among wines. The concept of perceived quality is even more critical when the studies are related to typical wine such as a Protect Designation of Origin (PDO).

In fact, in a PDO context or when describing a wine belonging to a particular region and produced with a recognizable grape variety, the intrinsic quality of wines could be summarized by the typicality assessment. Typicality is defined as the characteristics of a product from a terroir, meaning that the product is representative of its terroir, with terroir defined by two dimensions such as the environmental factors and the variety, cultivation, and winemaking practices [27]. Among them, the effect of interactions between the natural environment (soil and climate) and the vegetal material is known to be a major driver of wine typicality and quality [28,29,30,31,32]. Thus, typicality can be defined as a set of properties of belonging and distinction [33]. Considering the absence of defects as a pre-requisite, some authors [34,35] proposed that the intrinsic quality is the resultant of three different profiles: an eligibility profile, whose parameters are common to all wines (e.g., the sensory attributes and chemical compounds related to acidity, astringency, persistence, alcohol, viscosity, etc.); an identity profile (typicality), whose parameters are related to the grape variety and the terroir; a style profile related to the brand, expression of the kind of winemaking. The eligibility profile can change over time without affecting the identity of the wine [35], while the identity profile cannot change because it represents the distinct characteristics that define the typicality of a wine [36]. Finally, the style profile can chance overtime as function of the market or the winery brand needs, without altering the identity profile of the wine.

It is important to highlight that even in countries with consolidated protected designation of origin (PDO) and protected geographical indication (PGI) systems, a scientific approach of typicality still represents a challenge in terms of concept and sensory methodologies [26].

Since Sangiovese is one of the most wide-spread Italian red grapes and a very terroir-linked variety, it is important to study this grape cultivar on a broader scale in order to answer some key questions such as (i) the relationship between the chemical differences, in terms of eligibility and identity profiles, of the wines from different countries and the sensory profiles of the relevant wines, and (ii) how do Tuscan experts perceive the typicality of the Sangiovese wines from California and Italy, and which sensory descriptors best describe the wines.

The purpose of this study was to evaluate the intrinsic and perceived quality of Sangiovese wines from Italy and California, defining both similarities among wines from the same region of origin and the main differences between wines from Italy and California.

## 2. Materials and Methods

### 2.1. Sangiovese Wine Samples

Twenty commercial wines from the 2017 harvest (9 from Italy and 11 from California) were collected to be representative of both regions (Table 1). The selection of the wines was made according to the study of the 2016 vintage by the same authors [9]. All of the wines used in this study were sourced from commercial producers, were required to be 100% Sangiovese, and were not oak barrel aged. All wines selected did not show off-flavors. A minimum of 6 bottles were received for each wine sample. As the wines were made solely from Sangiovese grapes under commercial winemaking conditions, the differences in composition should reflect the regional styles. In California, the wines were chosen from the following American Viticultural Areas (AVAs): Central Coast, North Coast and Inland Valley region. The AVAs incorporate the following counties/regions: Amador County, Napa Valley, Santa Ynez Valley, Paso Robles, Saint Joaquin valley and Alameda County. In Italy, the wines were chosen from Tuscany representing the wine areas of Chianti Classico, Chianti, and Montalcino. As all wines used in this study were commercial products; there was no control over the viticultural or winemaking practices, and all participating wineries were asked for a “best representation of Sangiovese.

### 2.2. Intrinsic Quality: Chemical Characteristics for Measuring Eligibility, Identity, and Style Wine Properties

The eligibility chemical characteristics were represented by standard parameters, color indices, and polyphenol composition; the identity chemical characteristics were represented by the volatile fractions of the wines [34,35]. The style requirement was represented by the chemical variables related to wine aging (i.e., vanillin and γ-lactones).

The standard parameters (pH, titratable acidity, alcohol content, volatile acidity, residual sugar, malic acid) were measured with FT-IR analyses according to OIV/OENO Resolution 390/2010 and carried out by means of a FOSS WineScan (FT 120 Reference Manual, Foss, Hamburg, Germany).

Color intensity and hue were measured according to the method of Glories [37] and the total phenols index as described by Ribereau-Gayon [38]. The ultraviolet-visible (UV–vis) absorbance of the samples was measured on an Agilent spectrophotometer Cary 8454 UV–visible diode array detector, and the software used was UV–Visible ChemStation (Agilent Technologies, Inc., Little Falls, DE, USA). Milli-Q water was used as a reference (>18 MΩ·cm, Milli-Q Element system, Millipore, Bedford, MA, USA).

The polyphenol contents of the wines was determined according to Girardello et al. [39] methods using an Agilent (Santa Clara, CA, USA) 1260 Infinity HPLC equipped with an autosampler, temperature-controlled column compartment, and a diode array detector. The polyphenol composition of the wines is shown in Appendix A.

Free volatile compounds were determined according to the method developed previously [40] by HS-SPME GCMS. The analytical system for the determination was a Gerstel MPS2 autosampler (Gerstel, Baltimore, MD, USA) mounted to an Agilent 6890N gas chromatograph (Little Falls, DE, USA) paired with an Agilent 5975 mass-selective detector constituted the analytical system for the GC-MS analysis. The software used was MSD ChemStation (G1701-90057, Agilent Technologies, Inc., Little Falls, DE, USA). The volatile composition of the wines is shown in Appendix A.

All the wine samples were analyzed in triplicates for all chemical parameters.

### 2.3. Intrinsic Quality: Sensory Attributes Measuring Eligibility, Identity, and Style Wine Properties

#### 2.3.1. Selection of Samples

The sensory profiles of 12 out of the 20 Sangiovese wines, 6 of Italian origin (1I–6I) and 6 of American origin (7C–12C), 2017 vintage, were analyzed after 5 months of bottling. The selection of the 12 samples was carried over considering a malic acid concentration less than 0.5 g/L, and their representativeness of the different groups of wines defined by the chemical parameters.

#### 2.3.2. Training

The descriptive sensory analysis (DA) panel took place in the J. Lohr Sensory Room at the Department of Viticulture and Enology, University of California Davis. The panel consisted of 11 judges (8 females and 3 male). We recruited the participants from students, staff, and friends of UC Davis based on availability and interest. The protocol was exempt by the internal regulatory board (IRB).

The panel leader trained the panelists in six 60-min training sessions. In the first sessions, the panelists were presented with a range of Sangiovese wines and invited to describe them, generating descriptors and idea of standards. In the subsequent sessions, the judges were provided of subset of samples and reference standards until they reached the consensus about the attributes and the score sheet sequencing. The references were prepared from food and household products commonly available in the supermarket. The level of training of the panelists was checked by an individual evaluation of a subset of the samples and the analysis of the data.

#### 2.3.3. Evaluation of Samples

The eligibility sensory profile was described by the following 6 attributes: Sour, Sweet, Bitter, Viscous, Astringent, and Hot/Burning. The identity profile was defined by the following 13 aromatic attributes: Citrus, Floral, Black pepper, Dried fruit, Barnyard, Dark Fruit, Rubber, Cherry, Alcohol, Honey, Red berries, Earthy, and Bell pepper in-mouth flavor. One attribute described the style requirement, namely Oak odor and Oak in-mouth flavor.

The wine samples were served at room temperature in black glasses covered with plastic lids. Each sample contained a constant volume of 40 mL of wine. Wines were tasted blind, coded with randomized three-digit numbers, and served twice during the training period. At the end of the training period, the panel had consensus on 14 aromas, 3 taste and 3 mouthfeel descriptors for the wines (Table 2). The panelists had the opportunity to familiarize themselves with the data entry system FIZZ network (version 2.47B, Biosystèmes, Couternon, France) for collecting the attribute ratings.

Before each evaluation, panelists took an aroma quiz on the consent aromas, and on the first data collection day, panelists tasted the taste and mouthfeel standards.

The experimental design consisted of 12 wines served across judges in a balanced block design in triplicate. We labeled the wine samples with different 3-digit codes for each panelist. Panelists evaluated all wines in a total of 6 sessions, where one session consisted of 6 samples. Panelists rated the wines in individual, ventilated, and light isolating tasting booths under white light. For attribute ratings, we used an unstructured 15-cm line scale anchored by the wording “*not present*” to “*very intense*”, except for viscous, where the wines were rated from “*thin*” to “*thick*”. Judges were required to spit all the wine samples and wait 30 s between samples to clean their palates with water and unsalted crackers. At the end of each session, panelists had snacks, and after the study, panelists received a gift card. All the training and evaluation sessions were performed in about two months (1 or 2 sessions per week).

### 2.4. Perceived Quality: Napping^®^ Test, Wine Rating of Color and Typicality

Eleven Italian experts (3 females and 9 males; enologists, wine-producers, wine researchers), working with Sangiovese wine, with extensive experience (at least 5 years) with the different expression of Sangiovese wine produced in most important wine producing Italian areas, were involved in a sensory evaluation session in which three tests were performed: Napping*^®^* test and the typicality and color assessments. The same 12 wine samples submitted to the descriptive analysis were evaluated except for the sample 12C that was damaged during shipment from California to Italy.

These sensory evaluations took place in the wine sensory laboratory at the Department DAGRI, University of Florence (Florence, Italy).

The expert panelists were informed at the beginning of the study that the wines were all Sangiovese of the 2017 vintage. They were then instructed to assess the color of the samples presented under a white light, in glasses covered with plastic lids, labeled with capital letters, from A to M. The panelists were instructed as follows [41]:

“Imagine that you wanted to explain to someone what a Sangiovese wine color is. To explain, you can suggest to this person to evaluate a wine. For each wine presented, you must answer the following question: Do you think that this wine is a good example or a bad example of what a Sangiovese wine color is?”

The score of each sample was assigned on a categorical scale, from 1 to 7, anchored at left to “*very bad color*” and on the right to “*excellent color*”.

After this, judges proceeded to the Napping and typicality assessments. The Napping [42] is a specific variant of Projective Mapping, a method originally proposed for applied sensory studies by Risvik et al. [43] to describe overall differences among samples.

In both the Napping and typicality sessions, panelists were presented with the 11 wine samples plus one replicate. The evaluations were performed in isolated, ventilated sensory booths under red lights, to eliminate bias attributed to color differences. Each wine sample consisted of 25 mL of wine at room temperature (20 °C) presented, in clear 190 mL standard tasting glasses (ISO-3591, 1977) covered with plastic lids, and labeled with different random three-digit codes for each panelist and session. For both tests (Napping and typicality), a complete randomized and balanced experimental design was followed for the presentation order.

In the Napping session, the 12 samples were simultaneously presented to the panelists, who were then required to project them on a two-dimensional space (blank piece of white paper with dimensions 60 × 90 cm), in a way that reflected their perceived sample differences, i.e., by placing samples perceived as similar close to each other, and samples perceived to be more different further apart.

The data from the Napping test were digitized by writing in a table, for each product, its X-coordinate and its Y-coordinate on the sheet. The origin was placed on the left bottom corner of the sheet.

After finishing the Napping test, a new set of the same samples, with a different presentation order was presented to the panelists who were instructed as follows [41]:

“Imagine that you wanted to explain to someone what a Sangiovese wine is. To explain, you can suggest to this person to taste a wine. For each wine presented, you must answer the following question: Do you think that this wine is a good example or a bad example of what a Sangiovese wine is?”

The score of every sample was assigned on a categorical scale, from 1 to 10, anchored at left to “*very bad example*” and on the right to “*excellent example*”.

In each step, the samples were evaluated globally (i.e., orthonasal aroma after swirling, plus retronasal aroma, taste, and mouthfeel after sipping). Water was provided as a palate cleanser.

### 2.5. Statistical Analyses

The chemical and sensory data of the wines were analyzed by multivariate analysis of variance (MANOVA), with the factors as the wines and replicates, and frequency distributions were analyzed by the Chi-square test; all statistical analyses were completed using Statgraphics Centurion (Ver.XV, StatPoint Technologies, Warrenton, VA, USA).

Principal component analysis (PCA) was performed using XLSTAT v. 2018.3 (Addinsoft, Paris, France); partial least square regression analysis (PLS) was performed using Unscrambler (V9.1, CAMO Process AS, Oslo, Norway).

A multiple factor analysis (MFA) [44,45] in which each subject of the Napping^®^ panel (experts) constitutes a group of two un-standardized variables was performed using XLSTAT. The typicality scores of the second table (11 columns) and the Color scores of the third table (11 columns) were considered as two sets of 11 + 11 supplementary variables: They do not intervene in the axes construction, but their correlation coefficients with the factors of MFA are calculated and represented as in a usual PCA.

Descriptive Analysis data were analyzed by MANOVA, to check overall differences among the products for aroma, taste, and mouthfeel terms. Following a three-way ANOVA (analysis of variance) with the factors wine, judge, and replicate as well as their two-way interactions, Fisher’s LSD (least significant difference) test was used to detect differences among wines for the separate attributes. In those cases, where the effect of the wine was significant, but one of the interaction terms included wine as a factor, a pseudo-mixed model was applied. Here, a new F-value was calculated with the mean sum of squares from the significant interaction as an error term for the factor wine. The significance level for all statistical tests was set to *p* < 0.05.

## 3. Results

### 3.1. Evaluation of the Intrinsic Chemical Quality of the Sangiovese Wines

The chemical eligibility profile describes components common to all wines while the chemical identity profile describes components that may distinguish the territorial identity of the product. Figure 1 and Figure 2 reported the inherent chemical characteristics that represent the eligibility and identity wine properties for all of the Sangiovese wine samples. The style chemical characteristics that represent the chemical compounds related to the wine aging were not considered, since no relevant volatile compounds (i.e., vanillin and γ-lactones) were detected in the wine samples as they were collected before the oak barrel aging.

The chemical eligibility profile of the wines was represented by the standard chemical parameters (pH, titratable acidity, alcohol content, volatile acidity, malic acid, and residual sugar), color indices (color intensity, hue, and total phenols index) and polyphenol composition.

The chemical data were elaborated using the PCA, and the results were reported in Figure 1a,b. Figure 1a shows the observations (wines) on a two-dimensional map. Figure 1b shows the correlation circle (below on axes F1 and F2) and the projections of the initial variables in the factors space. The first two dimensions accounted for 57.90% of the total variance. The first dimension (37.61% of explained variance) separated the wines between Italy (I) and California (C) based largely on the polyphenol composition. According to the squared cosines of the variables for the two dimensions, it was possible to determine that on the first dimension (F1), quercetin (0.902), gallic acid (0.809), myricetin (0.796), total phenols index (0.757), color intensity (0.730), polymeric phenols (0.660), hue (0.609), residual sugar (0.531), volatile acidity (0.502), and pH (0.478) were the variables well linked to this axis. On the second dimension (F2), samples were separated by anthocyanin (petunidin-3-*O*-glucoside (0.961), delfinidin-3-*O*-glucoside (0.774), peonidin-3-*O*-glucoside (0.489), malvidin-3-*O*-glucoside (0.516)), and quercetin-3-*O*-galactoside composition (0.451). According to the importance of the above mentioned variables, the Italian wines on the right side of the plot were characterized by the polymeric phenols, monomeric anthocyanins (peonidin-3-*O*-glucoside, petunidin-3-*O*-glucoside, delfinidin-3-*O*-glucoside), color intensity, and total phenols index. The Californian wines on the left side of the plot were instead characterized by hue, residual sugar, pH, volatile acidity, malvidin-3-*O*-glucoside, and quercetin. The two ellipses defined the interval of confidence (95%) and helped to evidence a better separation between wines in the two regions.

The chemical identity profile of the Sangiovese wines was represented by the volatile compounds originating in the grape and by the alcoholic and malolactic fermentations (terpenes, norisoprenoids, acetates, esters, acids, alcohols).

The data were elaborated using the PCA, and the results were reported in Figure 2a,b. Figure 2a shows the observations (wines) on a two-dimensional map. The first two dimensions accounted for 37.07% of the total variance. According to the squared cosines for the variables on the two axes (F1 and F2), the volatiles that were well linked to the first dimension were ethyl octanoate (0.880), octanoic acid (0.741), ethyl butanoate (0.709), ethyl decanoate (0.613), isoamyl acetate (0.554), 4-terpineol (0.460), and β-phenylethanol (0.410), while compounds on the second dimension were β-phenethyl acetate (0.736), TDN (0.659), 3-methylbutan-1-ol (0.605), isoamylbutanoate (0.590), and β-damascenone (0.522).

Wines were separated according to the region of origin along the first dimension. In particular, most of the Californian wines were located on the right side of the plot described by volatile compounds such as esters (i.e., ethyl hexanoate, ethyl butanoate, ethyl octanoate, ethyl decanoate), acetates (ethyl acetate, β-phenethyl acetate, ethyl lactate, diethyl succinate, 3-methylbutyl acetate) and fatty acids (decanoic acid, dodecanoic acid). On the left side of the plot, the Italian wines were characterized mostly by varietal volatile compounds such as terpenes β-citronellol, β-linalool, 4-terpineol, α-terpineol) and norisoprenoids (vitispirane I, riesling acetale, β-farnesene). However, the wines from the two different origins were not completely separated by the identity profile as evidenced by the overlapping confidence intervals (95%) of the two ellipses (Figure 2a).

### 3.2. Evaluation of the Intrinsic and Perceived Sensory Quality of the Sangiovese Wines

In order to understand if the chemical differences evidenced in the wines from Italy and from California led to sensory differences, two sensory different panels (trained judges and experts) evaluated the wines. The panel of trained judges defined the sensory descriptive profile (sensory intrinsic quality); the experts performed the Napping test and evaluated the typicality assessment and quality of color (sensory perceived quality).

#### 3.2.1. Evaluation of the Intrinsic Quality: Eligibility and Identity Sensory Profiles

Using Descriptive Analysis, the panel of trained judges described the sensory attributes of the Italian and Californian wines. Figure 3a,b shows the distribution of the wines according to the sensory eligibility (sensory attributes common to all wines) and identity descriptors (sensory attributes that distinguish the territorial identity of the product). Only significant descriptors were used for the PCA, and 79.30% of the total variance was explained by the first two factors/dimensions. The identity descriptors (in red) were organized along the first bisector and contrasted two groups of samples, the Italian wines 4I, 5I and to a lesser extent 6I, on the right side and all the Californian samples (7C, 9C, 8C, 10C, 11C, 12C) and the Italian samples 1I, 2I, 3I on the left side. The notes of Cherry, Red Berries, Citrus, Floral, and Honey characterize the samples of the left side, while Earthy, Barnyard, and Rubber of the two Italian wines were on the right side of the plot. The second dimension separates the samples according to the Bell Pepper attributes in the positive dimension (10C, 7C) as opposed to the attributes Honey, Cherry and Floral (2I, 3I, 12C) in the negative dimension. The eligibility attributes (in blue) separate the samples along the same axis with Sweet, Burning/Hot and Alcohol correlated to the left group wines (7C, 9C, 8C, 10C, 11C, 12C, 1I, 3I, 2I) and Sour and Astringent to the Italian wines on the right (4I, 5I).

#### 3.2.2. Evaluation of the Perceived Quality: Napping^®^ Test, Wine Rating of Color and Typicality

In order to correlate the results from the panel of experts and describe the correlation between the Napping test, typicality, and quality of color evaluation, the results were analyzed by multiple factor analysis (MFA).

The X- and Y- coordinates from the Napping test of each wine sample generated a consensus configuration for the wines based on the individual maps of the panelists (11 judges) (Figure 4a). Dimension 1 (F1) and 2 (F2) of the MFA represent 34.40% and 24.10% of the total variance (58.49%), respectively. According to their larger distance along the first dimension, it can be seen that the group of Californian wines were positioned on the left side of the graph and were perceived to be very different from the Italian wines on the right side. The sample 2I rep was from the replicates of the 2I wine, and it was the only Italian wine positioned on the fourth quadrant together with the Californians.

The data from the typicality and quality of color valuations were plotted on the Napping results as Appendix A [41] (Figure 4a,b). The color scores were characterized by a greater dispersion in the MFA plot (Figure 4b) than the typicality scores. The average typicality score for the Californian wines was higher than the average of the Italian wines (8.10 vs 8.00) but not significant; however, the overall highest typicality score was received for an Italian wine, 2I (score of 9.54 and 8.63 for the replicate) (Table 3).

### 3.3. Relationship between Identity Profile and Typicality of Wines

In order to understand how the typicality of Sangiovese could be described and related to sensory attributes and volatiles, partial least square regression analysis (PLS-1) was performed. Only chemical intrinsic parameters of the 12 wines selected were considered. Figure 5 shows the X- and Y-loadings plot of sensory attributes X (related to identity profile) versus typicality scores Y of the wines. The total explained variance was 69% PC1, 18% PC2 in the X-dimension, and 23% PC1, 14% PC2 in the Y-dimension. Honey, Cherry, Floral, Red Berries, and Citrus aroma attributes were positively correlated to typicality, while Barnyard, Earthy, Rubber, and Bell pepper were negatively correlated (Figure 5). In particular, the important variables in describing the model were Earthy, Cherry, and Red berries as evidenced from running the uncertainty test in the PLS-1 regression (Figure 5). Cross validation was applied. The predicted versus reference plot (Figure 6) shows that the predictions are high quality. The prediction error decreased significantly after eight PLS factors, and these represent the optimal model conditions (root mean square error (RMSE) cal = 0.0395 and RMSE val = 0.521); the overall correlation is satisfactory (R2 = 0.999).

The identity sensory attributes could be also related to the identity chemical characteristics of Sangiovese. Figure 7 shows the prediction of the identity sensory attributes (Y) by the volatile profile of wines (chemical identity) (X). The explained variance was 24% PC1, 17% PC2 in the X-dimension, and 48% PC1, 23% PC2 in the Y-dimension. According to the uncertainty test, the important variables for the prediction model were ethyl butanoate, isoamylacetate, ethyllactate, ethyloctanoate, ethyldecanoate, isoamyl octanoate, β-phenethyl acetate, β-damascenone, and octanoic acid. Taking into account that the important variables for the typicality prediction and with a positive correlation were Cherry and Red Berries, it is interesting to understand which chemical variables were particular related to these identity sensorial attributes.

The Red berries sensory attributes was very well predicted with a good R2 = 0.967 and RMSE = 0.172 on the PC 7. The chemical volatiles that described the Red berries attribute are the same that describe the model. Instead, for Cherry attribute, it is possible to create a good calibration model that did not have a good validation. The volatiles that better described the Cherry attribute were ethyl butanoate, isoamyl acetate, ethyl lactate, ethyl octanoate, ethyl decanoate, isoamyl octanoate, β-phenethyl acetate, β-damascenone, and octanoic acid. Finally, even if the Honey attribute was not important for the prediction of typicality of wines, it is interesting to evidence how it was very well associated to β-damascenone.

## 4. Discussion

One of the main objectives of this study was to examine intrinsic quality of the samples, evaluating how chemical differences in wines from Italy and California, in terms of eligibility (attributes common to all wines) and identity profiles (attributes that distinguish among different territorial identities), could reflect on the sensory perception of wines. In this context, the chemical eligibility profile of the wines was represented by the standard chemical parameters (pH, titratable acidity, alcohol content, volatile acidity, malic acid, and residual sugar), color indices (color intensity, hue and total phenols index) and polyphenols composition, while the chemical identity profile was represented by the volatile compounds originating in the grape and by the alcoholic and malolactic fermentations (terpenes, norisoprenoids, acetates, esters, acids, alcohols).

The experimental data showed that the Sangiovese wines from Italy and California resulted in differences mostly for chemical eligibility profile. In particular, it was very evident that the Italian and Californian wines differed in their color indices and polyphenol composition (eligibility). In fact, the Italian wines were higher in polyphenols compounds and in color intensity. These results were in agreement with the chemical characterization of Sangiovese wines from Italy and California for the 2016 harvest [9] where the Italian wines resulted in higher color intensity and total phenols index compared to the Californian ones, that showed instead a higher hue. The values for these indices were consistent with other findings for Sangiovese wines [8,46]. The differences in color indices were better explained by the polyphenols compounds of the wines that resulted in higher amounts for the Italian wines. They showed higher amounts of pigmented polymers and monomer anthocyanins (peonidin-3-*O*-glucoside, petunidin-3-*O*-glucoside, delphinidin-3-*O*-glucoside).

Sangiovese red grape is considered a variety with a neutral aroma since the total amount of terpenes is lower than 1 mg/L, and this variety is not dependent upon monoterpenes for its varietal flavor [47]. In the Sangiovese pulp and skin some norisoprenoids precursors such as TDN, riesling acetale, damascenone, and vitispirane [4] are detected. The above varietal volatile compounds were determined in both Italian and Californian wines indicating that the varietal aspects of the Sangiovese grape were maintained in both regions. Important differences were instead evidenced in wines from both regions according to the fermentative volatile compounds. The Californian wines were richer in the composition of these fermentative volatiles than the Italian wines confirming the trend observed for the same regions for the 2016 vintage [9]. Significant differences for volatiles between the two regions were reported in Appendix A.

Based on the chemical differences in the composition of the wines from the two regions, we further explore how intrinsic quality, in terms of chemical differences, could be reflected on eligibility and identity sensory profiles of the wines. Moving to the perceived quality, the second target of the study was to see how Tuscan wine experts perceived the peculiarity/typicality of the Sangiovese wines from Italy and California and to link the sensory descriptors that might be associated with the wines’ typicality.

The sensory profiles of Californian and Italian samples were separated from each other, and the analysis of the correlation with the descriptive attributes allows interpreting this separation in terms of differences of both eligibility and identity profiles. The Californian samples were correlated to the identity attributes Bell Pepper, Cherry, Red Berries, Citrus, Honey. The attributes Barnyard, Earthy and Rubber were correlated to 4I and 5I wines, while there was not any correlation with the Californian wines. Given that all the samples were checked for the defects before all the sensory tests, this contrast seemed to describe a freshness range, for which varietal aromas were perceivable in some samples (left side of Figure 3a), while in other they were hidden by some typical aromas of a full-developed wine (right side of Figure 3a). The eligibility profile underlines this separation, with the samples on the left side of Figure 3a correlated to Sweetness that, even if associated to Alcohol and Burning/Hot, elicits a softer sensation respect to the Astringency and Sour of the Italian ones on the right side.

The perceived quality was studied by the Napping and typicality test. The Napping test results showed that the Californian and Italian wines (globally evaluated except for the color) were clearly separated, evidencing that the two kind of wines were perceived differently for the gustative and olfactive characteristics.

Despite of that, the results of the typicality evaluation did not show the same clear discrimination between the wines from the two regions: The average score of all the wines were very similar with the Californian wines slightly higher but not significantly different (Table 3). Figure 4 showed the correlation between the distribution of the wine samples according to Napping X- and Y-coordinates: the subjects were positioned in the second, third and fourth quadrants of the graphic with a concentration of the higher scores in correspondence of the position of two Californian samples (7C, 11C) and one Italian (2I). This result can be explained by the fact that, even if the expert subjects perceived differences among the wines, they did not associate them uniquely to typicality. Given the extensive training and experience of the experts, the lack of agreement among them can be interpreted not only as a variability of their opinion [48] but also as an indication that from the point of view of perceived quality in terms of typicality of the Sangiovese wine, the experts viewed all of the wines as falling within the identity profile. At the same time, the distribution of the higher average scores denotes that typicality has been correlated to fruity and floral attributes, in opposition to Bell pepper, Barnyard, Rubber, and Earthy descriptors [49]. In other words, the typicality of Sangiovese has been connected to the perception of the varietal characteristics that in this wine were related overall to fruity and secondly to floral [8]. These findings were evidenced by the PLS prediction of the typicality by the identity sensory attributes such as Cherry and Red Berries.

In the case of color evaluation, the experts more clearly separated the wines (Figure 3) and overall the Italian wines had significantly higher scores. In fact, these samples, reflecting the chemical parameters, had a more intense color and overall a lower hue compared to the Californian ones.

These results showed that the Sangiovese variety is recognizable even if grown abroad, very far from the original terroir of Italy and in particular in Tuscany. This is supported by the fact that the varietal volatiles were found in both wines from both countries, even if the Californian wines were more intense in fermentative volatiles than Italian wines were. Despite this, the main differences seemed related more to the intrinsic quality in terms of eligibility chemical and sensory profiles. Important and significant differences were found in wines for the polyphenol composition since Italian wines were higher in color intensity, tannins, monomeric anthocyanins, and pigmented polymers content. Consequently, they were perceived more intense in color and astringency. On the other hand, Californian wines were higher in alcohol content and pH and lower in titratable acidity compared to the Italian wines. These results reflected the eligibility sensorial perception of the wines in which the Italian wines tend to be more acidic, less sweet, and more astringent than their Californian counterparts.

These results evidenced that the terroir seemed to influence the eligibility characteristics of the Sangiovese grape variety, in particular for the polyphenol composition. In Italy, wines with a designation of origin are subject to production requirements that dictate many aspects of wine production such as the maximum grape yields, alcohol level, irrigation, and other quality factors, before an appellation name may legally appear on a wine bottle label [50]. In general, the US has the highest national average yields, at 6.5 tons/acre (115 hL/ha) (according to OIV), and the only requirement to use the AVA name on the wine label is that 85% of the wine must have come from grapes grown within the geographical AVA boundaries.

This aspect could have an important influence in the eligibility characteristics such as polyphenol composition. In fact, the content of polyphenols in grapes is clearly affected by four agro-ecological factors: the cultivar, the year of production (i.e., the climatic condition from year to year), the site of production (the effect of geographic origin of grapes, soil chemistry, and fertilization), and the degree of maturation [51]. Moreover, the methods of vinification and applied technological procedures (maceration, fermentation, clarification, aging, etc.) can significantly modify both the concentration and composition of polyphenols and, therefore, also the color intensity and hue of red wines [51]. The increasing of yield and vine vigor can also affect the color, polyphenol composition, and sensory attributes of Cabernet Sauvignon wines [52,53].

In conclusion, Sangiovese wines from Italy and California showed several significant chemical differences in term of eligible and identity profiles (intrinsic quality), such as polyphenols composition and volatiles that not completely affected the intrinsic sensory quality. Concerning the perceived quality, despite the Tuscan experts perceived differences between the Californian and Italian wines, they considered them similar when they evaluated their typicality.

Finally, the results from this study confirm that perceived quality in terms of typicality of Sangiovese was still related not only to fruity and floral flavors but also to lightness and freshness, despite the intrinsic quality aspect of the “structure” of the wine and to what is considered a “good” color. Moreover, the findings confirm that Sangiovese shows a flexibility in terms of chemical and sensory modification, according to the production area and that it can be considered typical even if it comes from an area far away from the traditional ones.

## Figures and Tables

**Figure 1 foods-09-01088-f001:**
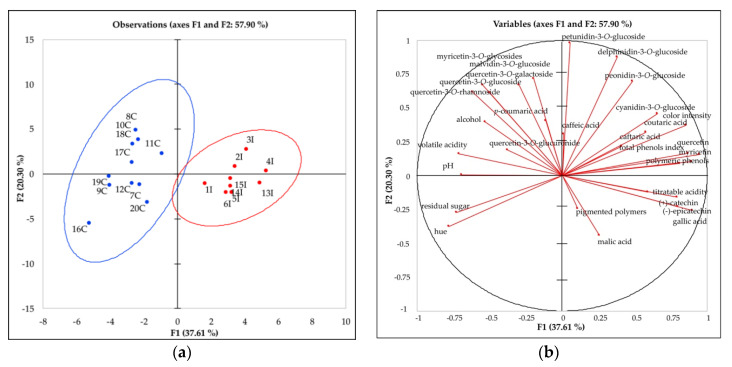
Principal component analysis (PCA) scores (**a**) and loadings (**b**) plots of eligibility profile (standard chemical parameters, color indices, and polyphenol compounds) for Sangiovese wines from Italy (in red) and California (in blue) from 2017 harvest. See Table 1 for the wine codes.

**Figure 2 foods-09-01088-f002:**
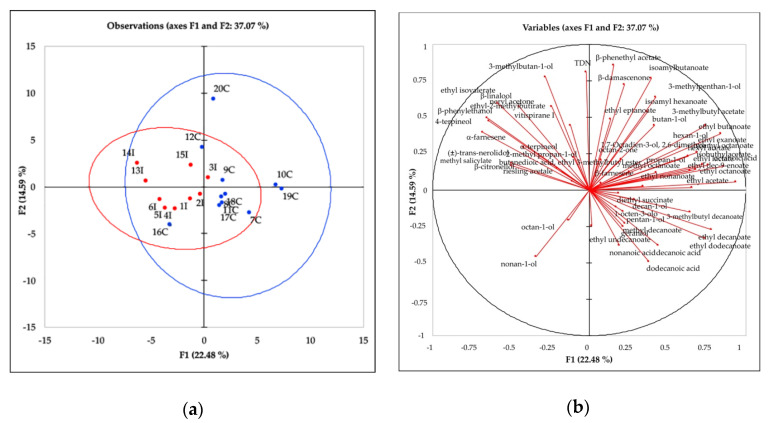
Principal component analysis (PCA) scores (**a**) and loadings (**b**) plots of identity profile (volatile compounds) for Sangiovese wines from Italy (in red) and California (in blue) from 2017 harvest. See Table 1 for the wine codes.

**Figure 3 foods-09-01088-f003:**
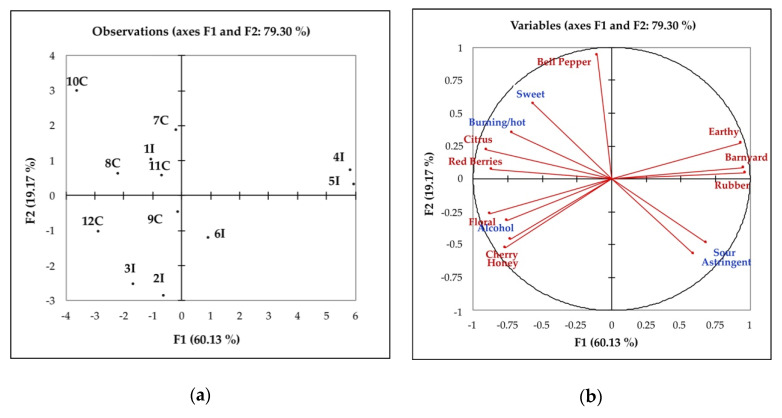
Principal component analysis (PCA) scores (**a**) and loadings (**b**) plots of eligibility (blue) and identity (red) profile (QDA sensory attributes) for Sangiovese wines from Italy and California from 2017 harvest. See Table 1 for the wine codes.

**Figure 4 foods-09-01088-f004:**
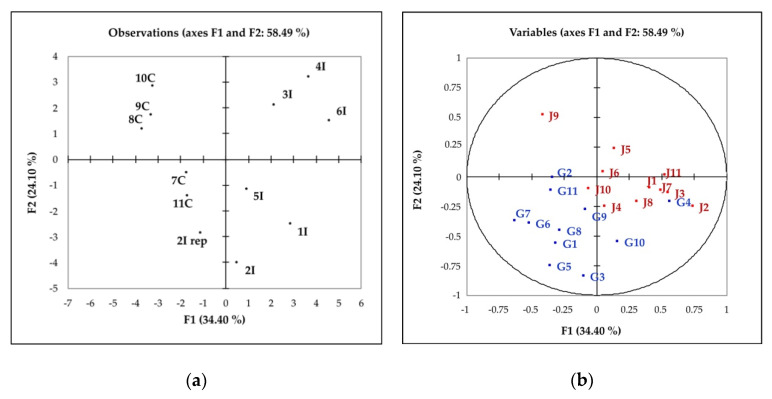
Representation of the Italian and Californian wines by multiple factor analysis according to the Napping X- and Y-coordinates, quality of color and typicality scores provided by the panel of experts. (**a**) Wines distribution (see Table 1 for wine codes); (**b**) distribution of the quality of color (j1–j11 in red) and typicality scores (G1–G11 in blue) (elaborated as Appendix A).

**Figure 5 foods-09-01088-f005:**
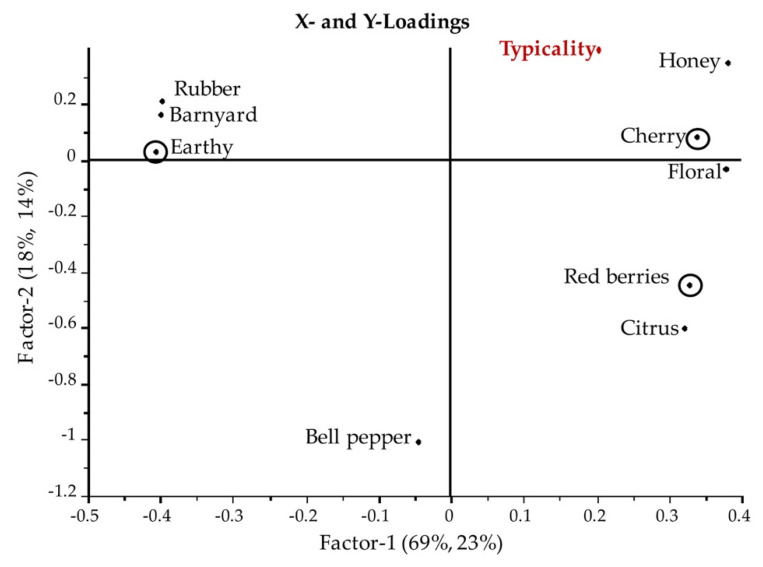
Partial least square regression (PLS-1) model for prediction of the typicality scores of wines by identity sensory attributes. Marked attributes with the circle around the dot were considered the important variables according to the uncertainty test (Earthy, Cherry, and Red berries).

**Figure 6 foods-09-01088-f006:**
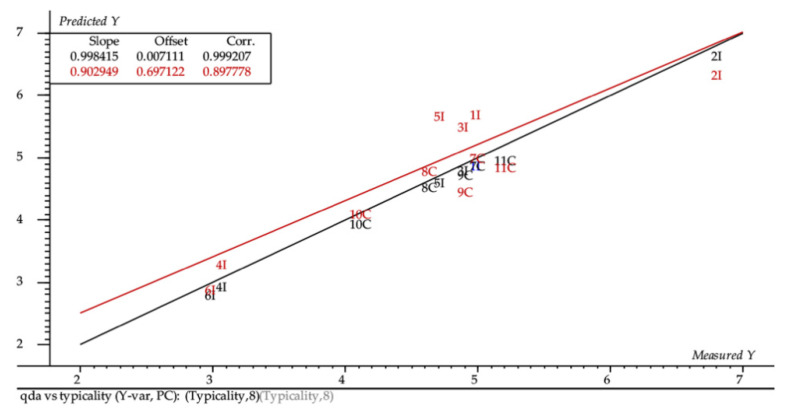
Comparison of predicted wine typicality plotted against the measured value (PLS-1). The black line is the reference model and the red line the predicted model. See Table 1 for wine codes.

**Figure 7 foods-09-01088-f007:**
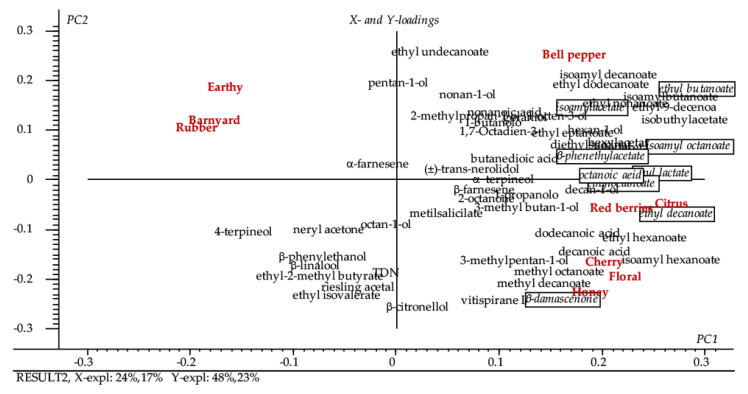
Partial least square regression (PLS-2) model for prediction of identity sensory attributes of wines by volatiles. Marked volatiles were considered the important variables according to the uncertainty test.

**Table 1 foods-09-01088-t001:** Details of the wines in the study from California and Italy, including the region of origin and the standard chemical parameters. Only the wines 1I–6I and 7C–12C were analyzed for the sensory tests.

Wine Code	State/ Country	Region	Alcohol (*v*/*v*%)	Residual Sugar (g/L)	Titratable Acidity (g/L)	Volatile Acidity (g/L)	pH	Malic Acid (g/L)
1I	Italy	Chianti Classico (Tuscany)	13.65	<1	5.36	0.48	3.55	0.00
2I	Italy	Chianti Classico (Tuscany)	13.30	<1	6.32	0.39	3.38	0.00
3I	Italy	Chianti Classico (Tuscany)	14.26	<1	5.85	0.29	3.45	0.00
4I	Italy	Chianti Classico (Tuscany)	12.65	<1	5.80	0.42	3.31	0.00
5I	Italy	Chianti Classico (Tuscany)	13.85	1	7.60	0.35	3.24	0.17
6I	Italy	Chianti Classico (Tuscany)	14.04	1.12	5.18	0.43	3.42	0.19
13I	Italy	Chianti (Tuscany)	14.23	<1	6.48	0.52	3.31	0.85
14I	Italy	Chianti (Tuscany)	14.05	2.12	6.37	0.37	3.29	0.93
15I	Italy	Montalcino (Tuscany)	14.45	0.23	6.82	0.21	3.43	0.90
7C	California	Napa Valley (North Coast)	13.32	2.59	4.60	0.54	3.72	0.09
8C	California	Santa Ynez valley (Central Coast)	15.18	3.56	5.00	0.66	3.58	0.08
9C	California	Napa Valley (North Coast)	14.17	3.27	6.03	0.72	3.59	0.00
10C	California	Alameda County (Central Coast)	15.38	2.78	4.76	0.59	3.68	0.00
11C	California	Napa Valley (North Coast)	14.90	1.88	4.13	0.55	3.79	0.09
12C	California	Amador County (Sierra Foothills)	13.88	2.64	5.48	0.53	3.43	0.00
16C	California	Saint Joacquin valley (Inland Valley)	14.67	8.37	3.58	0.74	4.01	0.33
17C	California	Napa Valley (North Coast)	14.42	2.30	4.33	0.73	4.00	0.49
18C	California	Santa Ynez valley (Central Coast)	15.39	3.16	5.98	0.64	3.38	0.13
19C	California	Paso Robles (Central Coast)	14.61	1.83	4.96	0.53	3.60	0.00
20C	California	Amador County (Sierra Foothills)	14.58	2.89	6.77	0.26	3.42	1.27

**Table 2 foods-09-01088-t002:** Complete list of descriptors and preparation.

Taste and Mouthfeel	Recipe	Product
Sour	2 g/L citric acid	Millard citric acid
Sweet	12 g/L cane sugar	C&H
Bitter	1.6 g/L caffeine	Sigma & Aldrich, food grade
Viscous	4 g/L CMC	Sigma & Aldrich, food grade
Astringent	1 g/L alum	McCormick
Hot/Burning	250 mL vodka/L	Smirnoffs
**Aroma**	**Recipe**	**Product**
Citrus	1 quarter whole fresh grapefruit, and a quarter of a whole fresh lemon	fresh produce, ruby red grapefruit, lemon
Floral	8 drops rose water, 8 drops orange blossom water and 8 drops violet syrup in 15 mL wine *	Sadaf Orange blossom water, Carlo Rose Water, Monin Violet syrup
Black pepper	1/8 teaspoon ground black pepper	Morton & Bassett, San Francisco Whole black pepper
Dried fruit	1 tsp dried figs, 1 tsp dried apricots (non-sulfured) and 1 tsp raisins	Sunsweet dates (chopped); Sunmaid raisins, sun-dried; unsulfured, Organic Malatya Apricots, by Mariani
Barnyard	1/8 tsp of ground white pepper in 30 mL of water	Morton & Bassett, San Francisco Whole White Pepper
Dark Fruit	5 mL Prune juice, 5 mL blackcurrant juice and 1 tablespoon blackberry jam	Sunsweet^TM^ Ribena black currant juice; Bon mamman blackberry jam
Rubber	1/2 new rubber bike hose in small pieces	Target brand
Cherry	1 tsp red tart cherry juice, 4 cherries	Red tart cherries in water, Oregon specialty fruits
Alcohol	20 mL of vodka and 10 mL of wine *	Smirnoff, Vodka
Honey	2 tsp raw honey	Nature Nate’s 100% Raw & Unfiltered Honey
Oak	2 small pieces of American oak, 10 drops of water	Evoak, large chips, heavy roast
Red berries	1 fresh strawberries, 2 fresh raspberries	Driscolls fresh fruits
Earthy	1 tablespoon potting soil, 5 mL water and 5 mL wine *	Miracle Gro Potting Mix
Bell pepper	5 g green bell pepper and 2 green beans	Fresh green bean and bell pepper purchased locally

* wine = Franzia Merlot.

**Table 3 foods-09-01088-t003:** Typicality and Color scores of the Italian and Californian wines assigned by the experts’ panel (mean values). See Table 1 for wine code, 2I rep is the replicates of the 2I wine ^1^.

Wine	Typicality Score	Color Score
1I	8.36 ab	3.45 d
2I	9.54 a	4.00 bcd
2I (replicate)	8.63 ab	4.73 ab
3I	8.09 bc	4.82 a
4I	6.73 cd	4.18 abcd
5I	8.00 bcd	4.18 abcd
6I	6.64 d	4.27 abcd
7C	8.36 ab	3.54 d
8C	8.27 ab	4.36 abc
9C	7.91 bcd	2.18 e
10C	7.45 bcd	3.54 d
11C	8.54 ab	3.91 cd
***F-value sample***	*2.51*	*5.72*
***p-value sample***	*0.0074*	*0.0000*
***Standard error sample***	*0.51*	*0.29*
***Average Italian***	*8.00 a*	*4.23 b*
***Average Californian***	*8.11 a*	*3.51 a*
***F-value region***	*0.93*	*5.27*
***p-value region***	*0.3370*	*0.0234*

^1^ Different letters within the same row indicate significant differences.

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
