# Peer review of "Evaluation of the Intrinsic and Perceived Quality of Sangiovese Wines from California and Italy"

_foods, 2020, doi:10.3390/foods9081088_

Round 1
Reviewer 1 Report
The paper is original and the data interesting. However, there are few points inside the manuscript that should be clarified in order to improve scientific quality.
In general: samples (wine, grapes, extracts…) are analysed and compounds are determined. For example, Line 128: pigments were determined or identified…
- In general: sometimes the numbering is with numbers and other times with letters, please unify throughout the manuscript. For example, Line 176: 6 sessions, where one session consisted of six samples.
- Nomenclature should be standardized, for example:
a)polyphenol profiles, polyphenols composition, polyphenol compounds, polyphenols content, polyphenol content. Why five different ways to name the same?
b)Line 128: monomeric phenolic and wine pigments. This line indicates “monomeric phenolic” and “wine pigments” were analysed but they are never named again. Why?
c)Line 278: “the polymeric pigments, monomeric anthocyanins”. Polymeric are pigments and monomeric are anthocyanins? Polymeric pigments are not anthocyanins? Please, name the compounds appropriately.
d)Line 429 and 485: colored polymeric pigments content? Are there no colored pigments? Please, unify the nomenclature.
e)Lines 122-123: there are abbreviations that are not used in the manuscript (CI and TPI). Why hue and Hue?
f)Units should be standardized
- Abstract: some numerical information should be added. The objective has to be clarified.
- Line 52: check space between references.
- Line 142: the section 2.3 should be divided into subsections. Methodology should be described in more details.
- Lines 275-282: the compounds with O-glucoside and O-galactoside, O in italics.
- Conclusions must be improved and be closely related to the results and objectives
- Figure 2. image quality is poor.
- A table with information of phenolic composition could be added.
Author Response
Comments and Suggestions for Authors
The paper is original and the data interesting. However, there are few points inside the manuscript that should be clarified in order to improve scientific quality.
In general: samples (wine, grapes, extracts…) are analysed and compounds are determined. For example, Line 128: pigments were determined or identified…
We have deleted line 128 and we have made corrections through the manuscript as suggested.
1.In general: sometimes the numbering is with numbers and other times with letters, please unify throughout the manuscript. For example, Line 176: 6 sessions, where one session consisted of six samples.
2.Nomenclature should be standardized, for example:
a)polyphenol profiles, polyphenols composition, polyphenol compounds, polyphenols content, polyphenol content. Why five different ways to name the same?
We have revised the manuscript to unify the nomenclature and numbers/letters as suggested.
b)Line 128: monomeric phenolic and wine pigments. This line indicates “monomeric phenolic” and “wine pigments” were analysed but they are never named again. Why?
We have corrected it as suggested and we have added the Table A as supplemental material that showed the list of all the wine pigments determined in both Italian and Californian wines.
c) Line 278: “the polymeric pigments, monomeric anthocyanins”. Polymeric are pigments and monomeric are anthocyanins? Polymeric pigments are not anthocyanins? Please, name the compounds appropriately.
We have corrected it as suggested.
d)Line 429 and 485: colored polymeric pigments content? Are there no colored pigments? Please, unify the nomenclature.
We have corrected it as suggested. Colored polymeric pigments was changed in pigmented polymers. In fact, the hplc method that was used is similar to the one described by Peng et al. 2002 (Peng, Z., Iland, P. G., Oberholster, A., Sefton, M. A., & Waters, E. J. (2002). Analysis of pigmented polymers in red wine by reverse phase HPLC. Australian Journal of Grape and Wine Research, 8(1), 70-75.) and the authors define the pigmented polymers “obtained by adding excess Sulphur dioxide to a wine and measuring the remaining colour at 520nm spectrophotometrically”.
e)Lines 122-123: there are abbreviations that are not used in the manuscript (CI and TPI). Why hue and Hue?
As suggested, we have removed these abbreviations through the manuscript
f)Units should be standardized
We have looked through the manuscript to standardize the units.
4.Abstract: some numerical information should be added. The objective has to be clarified.
We have corrected it as suggested.
5.Line 52: check space between references.
We have corrected it as suggested.
6.Line 142: the section 2.3 should be divided into subsections. Methodology should be described in more details.
We have modified it as suggested.
7.Lines 275-282: the compounds with O-glucoside and O-galactoside, O in italics.
We have corrected it as suggested
8.Conclusions must be improved and be closely related to the results and objectives
We improved the conclusion section as suggested.
Figure 2. image quality is poor.
We improved the quality of figure 2.
9.A table with information of phenolic composition could be added.
As suggested, we have added a Table A as supplemental material.
Reviewer 2 Report
Foods manuscript 881078
Overall comment: This study explored the differences between the Sangiovese wines produced in Italy and California. The wines were charactered using quantitation of chemical compounds, sensory descriptive analysis, napping analysis, and expert assessment of ‘typicality’ and ‘colour’. This study presents a wealth of data. However, in my opinion, some of the statistical analyses can be better explained. Also, the stated research aim is on ‘intrinsic’ and ‘perceived’ quality (these two terms are also in the title), but these terms were not properly defined, and the parameters measured in the study were not properly linked to the quality assessment. In fact, ‘intrinsic’ and ‘perceived’ quality were completely omitted from discussion. In my opinion, the results and discussion of this paper needs to be extensively revised.
My detailed comments are as follows.
Line 61: The meaning of ‘intrinsic’ and ‘perceived’ quality should be explained so that the readers can have a clear understanding of the research question being addressed.
Line 73 – 79: Here, the authors quoted a theory which divided wine quality into three different profiles. Subsequently, certain wine parameters were measured because they reflected the profiles. How are these profiles related to ‘intrinsic’ and ‘perceived’ quality, which is the aim of the research?
Line 80: ‘effecting’ should be ‘affecting’.
Line 128 – 134: Authors reported two methods of measuring monomeric phenolic and wine pigments. This is confusing. Were they used to measure different wine components? If so, please provide more details.
Line 143: Authors states that only 12 out of the 20 wines finished MLF and/or were free of off-flavours. Does this mean that 8 samples either didn’t finish MLF and/or exhibited obvious fault? If so, are their chemical profiles representative of regional typicality? Especially considering the authors stated in line 72 that the absence of defects is a prerequisite for measuring wine quality?
Line 176: How far apart were the sessions spaced? Were they run within a day, a week, or over a longer period of time?
Line 299: I wouldn’t say that the wine were separated by the regions judged by how much the confidence interval overlaps. Did the authors look at which volatile compound exhibited significant difference between the two regions?
Figure 4: In my opinion, it is more informative to plot the loadings of colour and typicality scores on the napping XY plot to examine if they are associated with the sensory descriptions of the wines. The supplementary scores plot doesn’t offer much information. Authors stated that the colour scores had greater dispersion, but how was this quantified? Was it a mere observation from the scores plot?
Table 3: I think that the standard error should be reported. Were the p-values adjusted for bias?
Line 377: How was the model validated? Was it leave-one-out cross validation?
Line 388: 20 wines were used for chemical analyses and 12 for descriptive analyses. Is the PLS model calculated with data from the 12 wines used for DA? This should be stated.
Line 421: Judged by Figure 2, I don’t tend to agree that the regional differences can be characterised by the ‘identity profile’.
Line 441: ‘Italians’ should be ‘Italian wines’. Also, the concentrations of the volatile compounds were not presented. Was any statistical analyses performed to confirm that significant differences existed between the wines from the two countries in terms of any volatile compounds measured?
Line 452: ‘Italians’ should be ‘Italian wines’. I also have reservations about the statement that Italian wines had more pronounced ‘Barnyard’, ‘Earthy’, and ‘Rubber’ characters. It was only 4I and 5I that were highly correlated with these characters. And 4I had a very low typicality score, which means the expert did not recognise it as a good example of Sangiovese. In my opinion, the best can be said here is that the American samples were free of these sensory characters.
Line 463: Figure 3 should be Figure 4.
Line 471: The average typicality scores between Italian wines and Californian wines were not compared statistically. Are they significantly different? Seems like the average score for Italian wines was slightly lower only because 4I and 6I scored fairly low.
Line 508 to 512: This seems to be the conclusion of this study. However, it did not address the ‘intrinsic’ and ‘perceived’ quality which is the research aim. The relationships between the ‘profiles’ measured and the ‘intrinsic’/ ‘perceived’ quality are not explained clearly.
Author Response
Overall comment: This study explored the differences between the Sangiovese wines produced in Italy and California. The wines were charactered using quantitation of chemical compounds, sensory descriptive analysis, napping analysis, and expert assessment of ‘typicality’ and ‘colour’. This study presents a wealth of data. However, in my opinion, some of the statistical analyses can be better explained. Also, the stated research aim is on ‘intrinsic’ and ‘perceived’ quality (these two terms are also in the title), but these terms were not properly defined, and the parameters measured in the study were not properly linked to the quality assessment. In fact, ‘intrinsic’ and ‘perceived’ quality were completely omitted from discussion. In my opinion, the results and discussion of this paper needs to be extensively revised.
My detailed comments are as follows.
Line 61: The meaning of ‘intrinsic’ and ‘perceived’ quality should be explained so that the readers can have a clear understanding of the research question being addressed.
As suggested, we have modified this section as follow “Characterizing regional differences in wines requires studying the intrinsic quality as its inherent physical-chemical characteristics. Moreover, perceived quality is rated by experts, critics, and consumers, which altogether defines a profile that describes common regional characteristics among wines. The concept of perceived quality is even more critical when the studies are related to typical wine such as a Protect Designation of Origin (PDO).”
Line 73 – 79: Here, the authors quoted a theory which divided wine quality into three different profiles. Subsequently, certain wine parameters were measured because they reflected the profiles. How are these profiles related to ‘intrinsic’ and ‘perceived’ quality, which is the aim of the research?
We have revised this section in order to be clearer, as follow “Considering the absence of defects as a pre-requisite, some authors [33,34] proposed that the intrinsic quality is the resultant of three different profiles: an eligibility profile, whose parameters are common to all wines (e.g., the sensory attributes and chemical compounds related to acidity, astringency, persistence, alcohol, viscosity etc.); an identity profile (typicality), whose parameters are related to the grape variety and the terroir; a style profile related to the brand, expression of the kind of winemaking. The eligibility profile can change over time without affecting the identity of the wine [34], while the identity profile can not change because represent the distinct characteristics that define the typicality of a wine [35]. Finally, the style profile can chance overtime as function of the market or the winery brand needs, without altering the identity profile of the wine.”
Line 80: ‘effecting’ should be ‘affecting’.
We have corrected it.
Line 128 – 134: Authors reported two methods of measuring monomeric phenolic and wine pigments. This is confusing. Were they used to measure different wine components? If so, please provide more details.
We have corrected it.
Line 143: Authors states that only 12 out of the 20 wines finished MLF and/or were free of off-flavours. Does this mean that 8 samples either didn’t finish MLF and/or exhibited obvious fault? If so, are their chemical profiles representative of regional typicality? Especially considering the authors stated in line 72 that the absence of defects is a prerequisite for measuring wine quality?
We have modified this part in order to be clearer. Line 101: we have added the sentence “All wines selected did not showed off-flavors.”. Line 143 we have defined the wines selection criteria as follow “The sensory profiles of 12 out of the 20 Sangiovese wines, 6 of Italian origin (1I-6I), and 6 of American origin (7C-12C), 2017 vintage, were analyzed after 5 months of bottling. The selection of the 12 samples was carried over considering a malic acid concentration less than 0.5 g/L, and their representativeness of the different groups of wines defined by the chemical parameters.”
Line 176: How far apart were the sessions spaced? Were they run within a day, a week, or over a longer period of time?
As suggested, we have added this information on the manuscript.
Line 299: I wouldn’t say that the wine were separated by the regions judged by how much the confidence interval overlaps. Did the authors look at which volatile compound exhibited significant difference between the two regions?
We have deleted line 299. We have checked which compounds exhibited significant difference between the two regions but we have not reported this information in the manuscript. However, we have added a supplemental Table B that shows the significant differences in volatiles between regions.
Figure 4: In my opinion, it is more informative to plot the loadings of colour and typicality scores on the napping XY plot to examine if they are associated with the sensory descriptions of the wines. The supplementary scores plot doesn’t offer much information. Authors stated that the colour scores had greater dispersion, but how was this quantified? Was it a mere observation from the scores plot?
Your question is interesting, and during the manuscript arrangement and data elaboration, we asked ourselves if it should have been more informative associate the sensory data (intrinsic and perceived quality) of the two different panels (trained judges and experts respectively), by MFA analysis. Actually, we opted for keeping separated the two groups of data and, to study the relationship between them, running the PLS analysis. By this way, we first analyzed the trained judges’ results evidencing the relationship between sensory attributes and wine samples. Then we explored the relationship between the Napping coordinates and the typicality and color scores provided by the experts’ panel. The PLS analysis allowed us to study and predict the level of typicality (perceived quality) as function of the identity sensory attributes (intrinsic quality), evidencing the important variables for the model. Finally, we also model the relationship between identity sensory descriptors and chemical volatiles.
Regarding the question about color and typicality dispersion, we made a mere observation of the scores plot considering the number of quadrants were the scores were distributed: typicality scores distributed in 1° and 4° quadrant; color in all the quadrants).
Table 3: I think that the standard error should be reported. Were the p-values adjusted for bias?
We have improved Table 3.
Line 377: How was the model validated? Was it leave-one-out cross validation?
We have added the information at line 375.
Line 388: 20 wines were used for chemical analyses and 12 for descriptive analyses. Is the PLS model calculated with data from the 12 wines used for DA? This should be stated.
We have added this information at line 369.
Line 421: Judged by Figure 2, I don’t tend to agree that the regional differences can be characterised by the ‘identity profile’.
We have corrected line 420-423 as suggested.
Line 441: ‘Italians’ should be ‘Italian wines’. Also, the concentrations of the volatile compounds were not presented. Was any statistical analyses performed to confirm that significant differences existed between the wines from the two countries in terms of any volatile compounds measured?
We have added Table B as supplemental with all this information.
Line 452: ‘Italians’ should be ‘Italian wines’. I also have reservations about the statement that Italian wines had more pronounced ‘Barnyard’, ‘Earthy’, and ‘Rubber’ characters. It was only 4I and 5I that were highly correlated with these characters. And 4I had a very low typicality score, which means the expert did not recognise it as a good example of Sangiovese. In my opinion, the best can be said here is that the American samples were free of these sensory characters.
We have corrected this part as suggested.
Line 463: Figure 3 should be Figure 4.
We have corrected it as suggested.
Line 471: The average typicality scores between Italian wines and Californian wines were not compared statistically. Are they significantly different? Seems like the average score for Italian wines was slightly lower only because 4I and 6I scored fairly low.
We have added the statistically comparison on table 3. We also have modified the manuscript according to the results and your suggestions.
Line 508 to 512: This seems to be the conclusion of this study. However, it did not address the ‘intrinsic’ and ‘perceived’ quality which is the research aim. The relationships between the ‘profiles’ measured and the ‘intrinsic’/ ‘perceived’ quality are not explained clearly.
We improved the conclusion section as suggested.
Round 2
Reviewer 1 Report
Line 194: change "were" to "was"
Congratulations for the work done!